# 1,2σ^3^λ^3^-Oxaphosphetanes and Their *P*-Chalcogenides—A Combined Experimental and Theoretical Study

**DOI:** 10.3390/molecules27103345

**Published:** 2022-05-23

**Authors:** Florian Gleim, Antonio García Alcaraz, Gregor Schnakenburg, Arturo Espinosa Ferao, Rainer Streubel

**Affiliations:** 1Institut für Anorganische Chemie, Rheinische Friedrich-Wilhelms-Universität Bonn, Gerhard-Domagk-Straße 1, 53121 Bonn, Germany; f.gleim@uni-bonn.de (F.G.); gregor.schnakenburg@uni-bonn.de (G.S.); 2Departamento de Química Orgánica, Facultad de Química, Campus de Espinardo, Universidad de Murcia, 30100 Murcia, Spain; antonio.garcia47@um.es

**Keywords:** phosphorous heterocycles, *P*-chalcogenides, strained molecules, oxidation, oxaphosphetanes

## Abstract

Although 1,2σ^5^λ^5^-oxaphosphetanes have been known for a long time, the “low-coordinate” 1,2σ^3^λ^3^-oxaphosphetanes have only been known since their first synthesis in 2018 via decomplexation. Apart from ligation of this P-heterocycle to gold(I)chloride and the oxidation using *ortho*-chloranil, nothing on their chemistry has been reported so far. Herein, we describe the synthesis of new 1,2σ^3^λ^3^-oxaphosphetane complexes (**3a**–**e**) and free derivatives (**4a**–**e**), as well as reactions of **4a** with chalcogens and/or chalcogen transfer reagents, which yielded the *P*-chalcogenides (**14**–**16a**; Ch = O, S, Se). We also report on the theoretical results of the reaction pathways of *C*-phenyl-substituted 1,2 σ^3^λ^3^-oxaphosphetanes and ring strain energies of 1,2σ^4^λ^5^-oxaphosphetane *P*-chalcogenides.

## 1. Introduction

Strained organic and inorganic ring systems [1] are of high interest, due to their special bonding situation and high reactivity; for example, oxetanes (**I**) (Figure 1) are important building blocks for the synthesis of more complicated molecules [2] and polymers [3]. The phosphorus-containing four-membered rings, the phosphetanes (**II**), drew attention because of their use as steering ligands in transition metal catalysis [4], and, more recently, their performance as organocatalyst [5,6,7]. The class of oxaphosphetanes can be regarded as an unusual combination of the features of oxetanes (**I**) and phosphetanes (**II**), which have been scarcely studied so far. Please note that isomeric 1,3-oxaphosphetanes (**III**) [8] and 1,2-oxaphosphetanes (**IV**) [9] do also exist.

In case of **III** and **IV**, the higher substituted compounds have been investigated more often. For example, 1,3σ^4^λ^5^-oxaphosphetanes are available through intramolecular Mitsunobu reactions [8], and bi- and tricyclic 1,3σ^3^λ^3^-oxaphosphetanes have recently been proposed in the decomposition of HPCO [10]. The high-coordinate 1,2σ^5^λ^5^-oxaphosphetanes (**IVb**) were known for a long time as intermediates in the Wittig reaction, although they do not occur in all cases [9,11,12], or in the deoxygenation of epoxides [12,13]. Recent calculations by Espinosa show that they also occur in the phosphite-initiated reductive dimerization of ketones [14]. Until now, very few crystal structures of 1,2σ^5^λ^5^-oxaphosphetanes (**IVb**) were reported [15,16,17,18].

In contrast, only the low-coordinate 1,2σ^3^λ^3^-oxaphosphetanes (**IVa**) were proposed [19] for a long time, with no stable derivative known. We synthesized the corresponding *κP*-pentacarbonylmetal(0) complexes (M = Cr, Mo, W) (**V**) either through ring expansion of epoxides using highly reactive phosphinidenoid complexes [20,21], or ring formation through intramolecular nucleophilic attack. Recently, the free ligand was obtained [22] using a decomplexation strategy [23] for octahedral complexes [M(CO)_5_L] by a combined thermal substitution with the chelating effect of bis(diphenylphosphino)ethane (DPPE). The first X-ray structure of a non-ligated 1,2σ^3^λ^3^-oxaphosphetane (**IVa**) was also reported together with the *P*-oxidation using *ortho*-chloranil and the *P*-complexation of gold(I)chloride [22].

Similar to 1,2σ^3^λ^3^-oxaphosphetanes (**IVa**), *P*-chalcogenides (**VI**) are rather elusive compounds. The 1,2-oxaphosphetane *P*-oxides (**VI**, E = O) were first reported by Regitz in 1973 [24], but it was later found by Inamoto that these products were in fact 3,4-dihydro-1*H*-2,3-benzoxaphosphorin 3-oxides [25]. In 1976, Inamoto proposed a 1,2-oxaphosphetane oxide as a reaction product of a phosphinidene oxide and *trans*-stilbene oxide [26]. However, the same author published a revised structure in 1991, showing that the product was in fact an acyclic secondary phosphine oxide [27]. In 1991, Hafez proposed an annulated 1,2-oxaphosphetane *P*-oxide-like structure for the photochemical reaction product of flavone with the Lawesson reagent. The product was only characterized by mass spectrometry, IR- and ^1^H-NMR spectroscopy, and elemental analysis, but as the publication lacks ^13^C- and ^31^P-NMR data and the four-membered ring bears no hydrogen atoms, this assignment might be incorrect [28]. In 1994, Okazaki reported the synthesis of a 1,2-oxaphosphetane *P*-oxide, kinetically stabilized through a bulky 2,4,6-triisopropylphenyl group at phosphorus [29]. Regarding the 1,2-oxphosphetane *P*-sulfides (**VI**, E = S), there is only one publication proposing 2-alkylthio-1,2-oxaphosphetane *P*-sulfides as the thermodynamically stable product of the reaction of 3-alkylamino-2-butenoic esters with phosphorus pentasulfide [30]. To the best of our knowledge, the 1,2-oxaphosphetane selenides and tellurides (**VI**, E = Se, Te) are unknown so far.

Herein, syntheses of new *C*^4^-substituted 1,2σ^3^λ^3^-oxaphosphetanes, the mechanistic evaluation of this reaction for model compounds using DFT calculations, as well as efforts to access their *P*-chalcogenides (**VI**, E = O, S, Se, Te) are described.

## 2. Results

### 2.1. Synthesis and Spectroscopic Characterization of 1,2σ^3^λ^3^-Oxaphosphetanes

Firstly, the protocol currently used for accessing 1,2-oxaphosphetanes [21,22,31] is significantly improved. In the absence of 12-crown-4 the *P*-triphenylmethyl (trityl) substituted Li/Cl phosphinidenoid complex **1** reacted cleanly with epoxides **2a**–**d** in THF yielding the oxaphosphetane complexes **3a,a’**–**3d,d’** (Figure 1). The new complexes **3b,b’**–**d,d’** could be isolated as pairs of diastereomers (Table 1). Compounds **3a,a’**–**d,d’** were then treated with 1,2-bis(diphenylphosphino)ethane (DPPE) at 80 °C for two days. The formation of the desired products **4a,a’**–**d,d’** was shown by ^31^P{^1^H}-NMR spectroscopy (Figure 1). For ^31^P{^1^H}-NMR parameters, as well as product ratios, see Table 2. For all **3b,b’**–**e,e’**, **3e*,e*’,** and **4b,b’**–**d,d’** diastereomeric pairs, the more highfield shifted ^31^P{^1^H}-NMR signal can be tentatively assigned to the *cis*-isomers and the downfield shifted signal to the *trans*-isomers, based on former calculations for **3a,a’** and **4a,a’ [22]**.

It should be noted that the change in isomer ratio from the complexes **3a,a’**–**d,d’** to the free 1,2-oxaphosphetanes **4a,a’**–**d,d’** can be attributed to the method of purification (extraction with *n*-pentane). At the end of the reaction, the isomer ratio closely resembles that of the starting material, the final difference arising from slightly different solubilities of the isomers in *n*-pentane.

Crystal structures of complexes **3b,b’**–**d,d’** were obtained (see ESI), but the change of *C*^4^-substituent did not lead to significant changes of bond lengths or angles compared to similar known compounds reported in the literature [21]. In the case of unligated species **4b,b’** (Figure 2) and **4c,c’** (see ESI), their crystal structures were obtained after recrystallization from *n*-pentane. The bond lengths of **4b,b’**–**c,c’** are very similar compared to their metal complexes **3b,b’**–**c,c’**. The bonds of phosphorus change by less than 2%. The change of the dihedral angle of the ring system is more prominent; for example, for **3b,b’** the dihedral angles are approximately 150° (*cis*) and 170° (*trans*), whereas the *trans* form of **4b,b’** is nearly planar and the *cis*-form bent more strongly (around 130°).

As in the case of the *P*-bis(trimethylsilyl)methyl substituted phosphinidenoid complex, the reaction of **1** with styrene oxide (**2e**) did not lead to the *C*^4^-, but preferentially to the *C*^3^-substituted 1,2-oxaphosphetane complexes [31]. The synthesis of a phenyl-substituted 1,2-oxaphosphetane was also attempted via reaction of **1** with the above mentioned oxirane derivative (**2e**), hoping to profit from the huge steric demand of the trityl group. However, this reaction led to a mixture of four isomers, the diastereomeric pairs of the *C*^4^- (**3e**,**e’**) and *C*^3^-substituted complexes (**3e***,**e*’**) (Figure 2), whose NMR data and ratios are collected in Table 3. The assignment of the ^31^P{^1^H}-NMR chemical shifts to the *C*^3^- and *C*^4^-substituted regioisomers is based on the *P*-bis(trimethylsilyl)methyl substituted case, where only the *C*^3^-substituted regioisomers are formed (proven by crystal structures), and where it is shown that they are downfield shifted in comparison to other *C*^4^-substituted derivatives [31]. Unfortunately, the mixture of **3e,e’** and **3e*,e*’** could not be separated using column chromatography, even at a lower temperature.

### 2.2. DFT-Based Mechanistic Proposal

Quantum chemical calculations were performed to provide further insights into mechanistic aspects of the formation of *C*-phenyl-substituted 1,2-oxaphosphetanes **3e,e’** and **3e*,e*’**. For the sake of computational economy, a methyl group (instead of trityl) was used as *P*-substituent in the Li/Cl phosphinidenoid moiety. Additionally, diethylene glycol dimethyl ether (DEGDME) was used as a model to provide an almost saturated coordination sphere for the Li cation. The approach of complex **5** to styrene oxide **2e** (taking the *S* enantiomer for the model study) gave rise to a van der Waals complex **6**, where the Li(DEGDME) group is coordinated to the epoxide O atom in a barrierless, thermodynamically favorable process, furnishing complex **7** (Figure 3). Given the high oxophilicity of phosphorus centers, nucleophilic attack of the negatively charged P atom to the electron-deficient O atom in the cationic part was first studied. By elimination of the solvated LiCl salt **8**, terminal phosphinidene-epoxide adduct **9** was formed in a markedly endergonic transformation, for which a TS could not be located. The singularity of a terminal phosphinidene pentacarbonyltungsten(0) oxirane adduct was recently studied [32], showing the weakest O→P bond among the whole series of cyclic ethers adducts of phosphinidene complexes. In contrast, complex **9** displayed a strengthened P→O bond with similar bond strength descriptors values than those obtained when the O donor is dimethyl ether **9^OMe2^** (Appendix A). Elongation of the less activated, non-benzylic C-O bond of **9** gives exergonically styrene **10** and phosphinidene oxide complex **11** through a moderate barrier (18.41 kcal mol^−1^). A similar result was found previously for a terminal phosphinidene molybdenum(0) thiirane complex, giving rise to ethylene and a *side-on* complexed phosphinidene sulfide [33]. On the contrary, P insertion into the benzylic C-O bond proceeds through a lower-energy TS (9.81 kcal mol^−1^) affording 1,2-oxaphosphetane **12*** (*C*^3^-substituted) in a markedly exergonic process.

A more favorable pathway to obtain the desired products resulted from the direct nucleophilic attack of the P atom to the epoxide C atoms of **7** (Figure 4). The attack at the more positively charged benzylic carbon (*q*^N^ = 0.03 e) is slightly kinetically favored (∆∆E^‡^_ZPE_ = 0.69 kcal mol^−1^) (Figure 3), due to a higher C-O bond activation (WBI = 0.881, MBO = 0.778) and the low steric hindrance of the methyl group at the phosphanido moiety. Conversely, the attack to the non-benzylic carbon atom (*q*^N^ = −0.10 e), with a comparatively strengthened C-O bond (WBI = 0.908, MBO = 0.945), leads to a more stable alkoxide **13** (Figure 3). However, when a *tert*-butyl group is attached to phosphorus, the attack to the non-benzylic carbon is (slightly) kinetically favored (see ESI). Therefore, in the real system with a trityl group, an even more favorable non-benzylic C-O insertion would be expected, due to the high steric hindrance. The cyclization to form the four-membered 1,2-oxaphosphetanes proceeds in both cases through similar energy TSs. The most stable isomer **12** is obtained (initially as the van der Waals complex **8·12**) through the slightly higher energy barrier process. The pathways leading to minor diastereomers **12′** and **12*’** were also computed (see ESI).

### 2.3. Synthesis of 1,2-Oxaphosphetane P-Chalcogenides

As the main goal of the study was to synthesize various 1,2σ^4^λ^5^-oxaphosphetane chalcogenide derivatives, the 42:58 mixture of 4-methyl-1,2-oxaphosphetane **4a,4a’** was used as a good case in point.

In order to target *P*-oxide derivatives, reactions of the mixture **4a,a’** with various oxygen-transfer reagents were studied. Treating **4a,a’** with propylene oxide or trimethylamine *N*-oxide in toluene at r.t. was not effective to convert **4a,a’** into **14a,a’**. The use of *tert*-butylhydroperoxide or *meta*-chloroperoxybenzoic acid (*m*CPBA) led to unselective reactions; however, the reaction using iodosylbenzene (Figure 5) led to the selective formation of 1,2-oxaphosphetane *P*-oxides **14a,a’**. The product was fully characterized by NMR spectroscopy, as well as ESI and APCI mass spectrometry. The ^31^P{^1^H}-NMR spectrum of the product solution showed two resonance signals of 62.1 ppm and 63.5 ppm, in a ratio of 66:34. This assignment fits well with the reported shift of the *P*-triisopropylphenyl substituted 1,2σ^4^λ^5^-oxaphosphetane *P*-oxide (δ (^31^P{^1^H}) = 48.7 ppm [29]).

To synthesize 1,2σ^4^λ^5^-oxaphosphetane *P*-sulfide **15a,a’, 4a,a’** was treated with elemental sulfur in toluene at ambient temperature (Figure 6). The reaction occurred selectively; **15a,a’** was isolated via extraction from *n*-pentane and it was fully characterized by NMR spectroscopy and LIFDI mass spectrometry. ^31^P{^1^H}-NMR chemical shifts of the isomers of **15a,a’** were observed at 115.8 (40%) and 120.0 ppm (60%). These values are close to those reported for a 2,5-dihydro-1,2-benzoxaphosphole-2-sulfide (130.2 ppm [34]).

Under the same reaction conditions, but with a slightly longer reaction time (2 d instead of 1 d), **4a,a’** was treated with elemental (gray) selenium. The 1,2-oxaphosphetane-*P*-selenides **16a,a’** (Figure 6) were formed in a selective manner and isolated as 29:71 mixture in good yields by filtration, and excess selenium was removed. The **16a,a’** mixture was fully characterized by NMR spectroscopy and LIFDI mass spectrometry. Its ^31^P{^1^H}-NMR spectrum showed two resonance signals with selenium satellites at 116.1 ppm (^1^*J*(Se,P) = 839.7 Hz) and 121.5 ppm (^1^*J*(Se,P) = 846.4 Hz), corresponding to the two diastereomers of **16a,a’**. The ^77^Se{^1^H}-NMR spectrum showed two doublets at −10.7 and 79.4 ppm. A comparison to the (acyclic) *tert*-butyl-ethoxyphenylphosphane-*P*-selenide[33] (δ (^31^P{^1^H}) = 111.0 ppm, ^1^*J*(Se,P) = 786.3 Hz) showed very similar values for the phosphorus chemical shifts and coupling constants, whereas the selenium resonances of **16a,a’** are downfield-shifted (cf. δ (^77^Se{^1^H}) = −350.3 ppm [35]).

However, **4a,a’** did not react with elemental tellurium or tributylphosphane-P-telluride (as transfer reagent) to form 1,2-oxaphosphetane-*P*-tellurides under the same conditions, nor by heating to 80 °C.

A comparison of the ^13^C{^1^H}-NMR data of **4a,a’**, **14a,a’**, **15a,a’,** and **16a,a’** (Table 4) reveals, for all chalcogenides, the ^1^*J*(P,C^H2^) and ^2^*J*(P,C^H^) coupling constants are increased as expected when oxidizing P(III) to P(V). In the case of **14a,a’**, the ^1^*J*(P,C^Ph3^) constant also increases, whereas it decreases when going from **4a,a’** to **15a,a’** and then to **16a,a’**. The decrease in the coupling constant hints at a change of the hybridization of phosphorus and, concomitantly, the bond angles due to increased steric bulk near the ring system.

### 2.4. Ring Strain Energy of Model 1,2-Oxaphosphetane Derivatives

To obtain further insight into the chemistry of the 1,2-oxaphosphirane chalcogenides, their ring strain energies (RSEs) were computed for model 1,2-oxaphosphetane derivatives **VIa-e** (Figure 7) using suitable homodesmotic reactions (see ESI), as previously completed for related three- and four-membered heterocycles [36,37,38,39,40,41,42,43,44]. RSEs values (Table 5) slightly increase in the order **VIa** < **VIb** < **VIc** < **VId** < **VIe**, therefore regularly increasing for heavier P-chalcogenides and also reproducing the reported variation on moving from the σ^3^λ^3^-1,2-oxaphosphetane **VIa** to its *P*-oxide derivative **VIb [22]**.

## 3. Materials and Methods

### 3.1. Synthetic Details

The syntheses of all compounds were performed under an argon atmosphere, using common Schlenk techniques and dry solvents. All NMR spectra were recorded on a Bruker AVI-300 or a Bruker AV III HD Prodigy 500 spectrometer at 25 °C. The ^1^H and ^13^C NMR spectra were referenced to the residual proton resonances and the ^13^C NMR signals of the deuterated solvents and ^31^P to 85% H_3_PO_4_ as external standards, respectively. Please check the ESI for further experimental details.

### 3.2. Computational Details

DFT calculations were performed with the ORCA electronic structure program package (version 4.2.1, created by Frank Neese, Max Planck Institut für Kohlenforschung, Mülheim/Ruhr, Germany) [45]. All geometry optimizations were run in redundant internal coordinates with tight convergence criteria, in the gas phase, and using Grimme’s dispersion-corrected composite PBEh-3c level [46]. For the mechanistic study, solvent (THF) effects were taken into consideration with the CPCM solvation method [47] as implemented in ORCA. For Mo [48] and Te [49] atoms, the [def2-ECP(28)] effective core potential (ECP) was used. Harmonic frequency calculations verified the nature of ground states or transition states (TS), having all positive frequencies or only one imaginary frequency, respectively. TS structures were confirmed by following the intrinsic reaction path in both directions of the negative eigenvector. From these optimized geometries, all reported data were obtained by means of single-point (SP) calculations using the more polarized def2-TZVPP basis set [50]. Reported energies include the Zero-point energy (ZPE) correction term at the optimization level. In the case of mechanistic aspects, final energies were obtained by means of double-hybrid-*meta*-GGA functional PWPB95 [51,52], using the RI [53,54,55] approximation for the MP2 correlation part, together with the RI-JK approximation for Coulomb and exchange integrals in the DFT part. Additionally, the latest Grimme’s semiempirical atom-pair-wise London dispersion correction D4 was included [56]. For RSE calculations, final energies were computed with the near-linear scaling domain-based local pair natural orbital (DLPNO) [57] method, to achieve coupled cluster theory with single-double and perturbative triple excitations (CCSD(T)) [58] using the def2-TZVPP basis set.

## 4. Conclusions

The new unligated 1,2σ^3^λ^3^-oxaphosphetanes **4a,a’**–**d,d’** were synthesized and characterized. As a good case in point, a mixture of **4a,a’** was used to investigate oxidation reactions, i.e., to access less (Ch = O) and/or unknown (Ch = S, Se) 1,2σ^4^λ^5^-oxaphosphetane *P*-chalcogenides. DFT calculations provided mechanistic insights into the formation of *C*-phenyl-substituted 1,2σ^3^λ^3^-oxaphosphetanes **3e,e’ and 3e*,e*’** using model derivatives. Nucleophilic attack to non-benzylic carbon of styrene oxide **2e** followed by cyclization seems to be the preferred pathway which explains the preferred formation of **3e,e’**. Ring strain energy calculations revealed the tendency to increase RSE values in going from 1,2σ^3^λ^3^- to 1,2σ^4^λ^5^-oxaphosphetane *P*-chalcogenides and among the latter from the lighter to the heavier chalcogens.

## Data Availability

The data presented in this study are available in Appendix A.

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
