# Peer review of "1,2σ3λ3-Oxaphosphetanes and Their P-Chalcogenides—A Combined Experimental and Theoretical Study"

_molecules, 2022, doi:10.3390/molecules27103345_

Round 1

Reviewer 1 Report

I recommend the following consideration for the publication.

  1. I recommend to submit the checkCIF files. How the purity of the complexes have been checked?
  2. Please check the complex 1 in the manuscript (Scheme 1, 2) as well as in ESI for the preparation of 3a,a’-d,d’.
  3. Please check the second temperature -60oC which may be   -50oC in the synthesis of complex 3a,a’ in the ESI file.
  4. I was wondered to identify the isomers from the NMR data only. I suggest to proceed further detail about the kind of isomers identification.
  5. I only found the crystal structure detail of 4b-c in manuscript and there is no any discussion about the 3b,c,d.
  6. Like Page 4, line 118, I suggest to keep 3e/3e’ and 3e*/3e*’ in scheme 2.  
  7. In the results section, I suggest to break down the section such as the syntheses scheme, crystal structures descriptions, quantum chemical calculations etc as per need.
  8. I also suggest to authors to keep the high quality pictures in figures.
  9. Under the materials and methods title, there is only computational details. I suggest to the authors to keep the appropriate details.   
  10. In ESI file, I found missing 4 a,a’ in content – “General procedures….”.
  11. I suggest to write the space group for complex 3c, 3d  P21/n  instead of author written  in ESI.

Author Response

see pdf

Reviewer 2 Report

In my opinion your manuscript should be prepared inits present form

Author Response

see pdf

Reviewer 3 Report

Prof. A. Ferao and prof. R. Streuble and co-workers reported the synthesis and structural analysis of novel 1,2s3l3-oxaphosphetanes, which are rare examples. Furthermore, their formation mechanisms were clarified by quantum chemical calculations. Synthesis of free 1,2s3l3-oxaphosphetanes has provided their P-chalcogenides for the first time. Using the model compounds of those chalcogenides, the strain energies of the four-membered ring compounds were calculated. This report contributes to the development of not only main group chemistry but also organic chemistry by synthesizing compounds that are almost unprecedented due to recently developed synthetic methods and investigated their chemical properties by single crystal X-ray structural analysis, spectroscopic analysis, and quantum chemical calculations.

My evaluation is that the paper is publishable with minor scientific revisions.

  • Previously, it was reported that decomplexation of 1,2s3l3-oxaphosphetane metal complexes did not change the isomer ratio of the product. However, this paper reports that the isomer ratio of the products changed from their metal complexes. If you have information on this, please mention it in the main text.

  • In ref 8, please add doi: 1055/s-0030-1260170.
  • ref 18, doi: 10.1021/ja00749a044
  • In ref 38, page number and doi are missing.

Regarding X-ray analysis, "All non-hydrogens were refined anisotropically" is not suitable, not All, iso-non-hydrogen atoms are included in CIFs.

And the typing styles of space group are irregular in the Supporting Information, please correct them.

******

Author Response

see pdf

Reviewer 4 Report

The authors reported the synthesis of new 1,2σ3λ3-oxa-phosphetane complexes of molybdenum(0) 3 and unligated ones 4 as well as the transformation of 4 into P-chalcogenides. Moreover, they performed theoretical studies on the formation mechanism of C-phenyl-substituted 1,2-oxaphosphetanes and the ring strain energies of 1,2σ4λ5-oxaphosphetane P-chalcogenides. The obtained products including rare or unknown 1,2-oxaphosphetane chalcogenides are interesting, and the manuscript is well written. Therefore, this referee considers that this manuscript is acceptable for publication in Molecules after addressing the following points.

1) Page 3,

The reason for the change in the ratio of the products in the conversion reactions of 3a,a’-d,d’ to 4a,a’-d,d’ should be commented on in the revised text. For example, why was 4d’ formed selectively?

2) Page S3 in the Supporting Information,

It would be better to describe the preparation method or commercially availability of the starting molybdenum complex with the PCl2CPh3 ligand for the readers to reproduce the preparation of 1,2σ3λ3-oxa-phosphetane complexes.

3) Page S3 in the Supporting Information,

The 1H NMR data of 3a,a’ should be checked. The signals assigned to the CH3 group (1.60 ppm and 1.14 ppm) in isomers 1 and 2 might need to be replaced each other according to ref 1 [Chem. Commun. 2018, 54, 7123–7126].

4) In the Supporting Information,

HRMS data are missing for some new compounds (3b,b’, 3c,c’, 3d,d’, 4b,b’, and 4d,d’). Please add the data in the revised Supporting Information.

5) In the Supporting Information,

The copies of 1H, 13C, and 31P NMR spectra of all new compounds should be provided in the revised Supporting Information.

Author Response

see pdf
